# The Mechanism of Important Components in Canine Fecal Microbiota Transplantation

**DOI:** 10.3390/vetsci9120695

**Published:** 2022-12-14

**Authors:** Kerong Li, Jie Yang, Xiaoxiao Zhou, Huan Wang, Yuxin Ren, Yunchuan Huang, Haifeng Liu, Zhijun Zhong, Guangneng Peng, Chengli Zheng, Ziyao Zhou

**Affiliations:** 1College of Veterinary Medicine, Sichuan Agricultural University, Chengdu 611130, China; 2Chengdu Center for Animal Disease Prevention and Control, Chengdu 610041, China; 3Sichuan Institute of Musk Deer Breeding, Chengdu 610016, China

**Keywords:** fecal microbiota transplantation, dog, efficient components, mechanism

## Abstract

**Simple Summary:**

Due to increasing global bacterial resistance, there is a growing demand to find alternative treatments to antibiotics. Fecal microbiota transplantation, which involves transplanting fecal material from a donor into a recipient’s gastrointestinal tract, is a possible alternative treatment for some diseases. However, which components are involved in canine fecal microbiota transplantation still need to be determined so that they can be better conserved when preparing fecal material. In this review, we focused on discussing the main components that play important roles in canine fecal microbiota transplantation and further interpreted how these components work.

**Abstract:**

Fecal microbiota transplantation (FMT) is a potential treatment for many intestinal diseases. In dogs, FMT has been shown to have positive regulation effects in treating *Clostridioides difficile* infection (CDI), inflammatory bowel disease (IBD), canine parvovirus (CPV) enteritis, acute diarrhea (AD), and acute hemorrhagic diarrhea syndrome (AHDS). FMT involves transplanting the functional components of a donor’s feces into the gastrointestinal tract of the recipient. The effective components of FMT not only include commensal bacteria, but also include viruses, fungi, bacterial metabolites, and immunoglobulin A (IgA) from the donor feces. By affecting microbiota and regulating host immunity, these components can help the recipient to restore their microbial community, improve their intestinal barrier, and induce anti-inflammation in their intestines, thereby affecting the development of diseases. In addition to the above components, mucin proteins and intestinal epithelial cells (IECs) may be functional ingredients in FMT as well. In addition to the abovementioned indications, FMT is also thought to be useful in treating some other diseases in dogs. Consequently, when preparing FMT fecal material, it is important to preserve the functional components involved. Meanwhile, appropriate fecal material delivery methods should be chosen according to the mechanisms these components act by in FMT.

## 1. Introduction

Fecal microbiota transplantation (FMT) is a process that involves infusing the fecal material from a healthy donor into a patient’s gastrointestinal tract. Through this process, the patient’s gastrointestinal microbial community becomes improved, which can alleviate gastrointestinal diseases. At present, FMT has been proven to have excellent effects on the treatment of recurrent *Clostridioides difficile* infection (CDI) in humans. In addition, it has also been successfully applied in the treatment of other gastrointestinal diseases, such as ulcerative colitis (UC), irritable bowel syndrome, idiopathic constipation, etc. [1,2]. We previously discussed the indications of FMT in dogs [3], illustrating that FMT has been found to positively regulate CDI [4], inflammatory bowel disease (IBD) [5,6,7], canine parvovirus (CPV) enteritis [8], acute diarrhea (AD) [9], and acute hemorrhagic diarrhea syndrome (AHDS) [10].

Generally, commensal bacteria are considered to be the main effective ingredients in FMT. However, viruses, bacterial fragments, proteins, antibacterial compounds, metabolites, oligonucleotides, and even cells shed from donors play important roles in human FMT, as Bojanova et al. [11] discussed. Ott et al. [12] also illustrated that delivering sterile fecal filtrates to CDI patients may help their intestines recover, which supports this viewpoint. Currently, studies on FMT in humans are more specific and thorough than those on FMT in dogs. Nevertheless, the intestinal microbiota of dogs are close to those of humans [13,14]. Accordingly, it is suspected that bacteria, viruses, bacterial fragments, fungi, mucin proteins, immunoglobulin A (IgA), and bacterial metabolites are also important components in canine FMT. In this review, referring to studies on FMT in humans and mice, we analyzed how the components in a canine donor’s feces induce intestinal recovery in the canine recipient after FMT.

## 2. Functional Components in Canine FMT

### 2.1. Microbiota

Gut microorganisms mainly contain archaea, fungi, bacteria, viruses, and protozoa. Bacteria are the most abundant microorganisms in feces and are also the most important components in FMT [15]. The main bacterial phyla in healthy canine feces include *Firmicutes, Bacteroidetes, Fusobacteria, Proteobacteria,* and *Actinobacteria* [16,17,18]. Meanwhile, the main bacterial genera are *Lactobacillus, Clostridium, Blautia, Dorea, Faecalibacterium,* and *Allobaculum*, which belong to the *Firmicutes* phylum; *Prevotella* and *Bacteroides*, which belong to the *Bacteroidetes* phylum; *Cetobacterium* and *Fusobacterium*, which belong to the *Fusobacteria* phylum; *Sutterella*, which belongs to the *Proteobacteria* phylum; and *Bifidobacterium* and other unclassified *Actinobacteria* [19]. The most abundant fungal phylum in canine feces is *Ascomycota*, including the classes *Saccharomycetes, Dothideomycetes, Pleosporales, Eurotiomycetes, Taphrinomycetes*, and so on [20,21]. Phages, such as the families *Siphoviridae* and *Myoviridae*, can be isolated from dogs [22,23].

#### 2.1.1. Commensal Bacteria

Commensal bacteria conserved from donor feces play an important role in FMT. In many studies, administering commensal bacterial strains showed a positive effect on treating various diseases. For example, *Bifidobacterium animalis* AHC7, isolated from the canine gastrointestinal tract, was found to significantly reduce *Clostridioides difficile* in canine gastrointestinal diseases [24]. Another study showed that the canine-derived *Bifidobacterium animalis* strain AHC7 may reduce the recovery time for acute idiopathic diarrhea in dogs [25]. Further, other probiotics in the intestine also play important roles in FMT, e.g., the *Lactobacillus acidophilus* strain LAB20 from the canine small intestine has been proven to have anti-inflammation and intestinal barrier functions [26,27]. The colonization of nontoxigenic *Clostridioides difficile* (NTCD) may prevent hamsters from dying from the *Clostridioides difficile* toxin [28,29]. Further, the administration of live *Faecalibacterium prausnitzii*, an abundant bacterium in *Firmicutes,* has been found to have a positive effect on protecting mice from dextran sodium sulfate (DSS)-induced colitis [30,31]. Oral *Akkermansia muciniphila* has been found to relieve DSS-induced colitis in mice as well [32]. The colonic administration of *Lactobacillus reuteri* has been found to protect rats from acetic acid-induced colitis [33]. The colonization of nontoxigenic *Bacteroides fragilis* in mice has been found to diminish colitis caused by enterotoxigenic *Bacteroides fragilis* [34]. Additionally, a study reported that introducing mixed commensal bacteria or probiotics into the intestine had positive effects on intestinal disease recovery [35]. Administering the probiotic VSL#3 (four strains of *Lactobacillus*, three strains of *Bifidobacterium*, and one strain of *Streptococcus sulivarius*) has been found to provide protection for dogs with IBD and aid in CPV therapy [36,37]. Some of these bacterial strains (probiotics or commensal bacteria) can be transferred from donor feces to recipients during FMT. Transferred commensal bacteria play their respective roles in the recipient’s gut, ultimately acting to affect diseases.

#### 2.1.2. Phages and Commensal Fungi

Phages are capable of killing specific bacteria, while they are unlikely to disturb normal flora [38,39]. A metagenomic study found that many temperate phages, which are useful to control invading pathogens and modulate microbial community structure, were transplanted during FMT, with *Siphoviridae* having the highest transfer efficiency [40]. Moodley et al. [22] isolated four bacteriophagic strains of *Siphoviridae* from canine feces, which had the ability to lyse the pathogen *Staphylococcus pseudintermedius.* Xue et al. [41] found that the oral *Yersinia* phage X1 in mice had a useful effect on controlling yersiniosis. The oral administration of bacteriophages has been found to reduce mortality in mice with gut *Pseudomonas aeruginosa*-derived sepsis [42]. The infusion of phage combinations in a hamster CDI model has been found to reduce *Clostridioides difficile* colonization and delay the appearance of symptoms [43,44].

Commensal fungi in healthy guts play a role in ameliorating intestinal diseases as well. *Saccharomyces cerevisiae* is the main component of the intestinal fungal community in mice. Research has found that *Saccharomyces cerevisiae* raises the survival rate of mice infected with *Salmonella enterica* serovar *Typhimurium* [45]. Further, the alleviation of clinical symptoms was observed in ulcerative colitis (UC) murine models treated with *Saccharomyces cerevisiae* [46,47]. In post-weaning pigs, feeding *Cyberlindnera jadinii* yeast was beneficial for gut homeostasis and made the pigs more robust [48].

### 2.2. Metabolites

The most potent metabolites in FMT are thought to be short-chain fatty acids (SCFA) or postbiotics. SCFAs are mainly produced in the colon and are mostly derived from carbohydrates in undigested food residue through anaerobic fermentation [49,50]. Studies have found that about 5–10% of SCFAs, including acetate, propionate, and butyrate, are not able to be absorbed in the intestine and thus remain in feces [50,51,52]. In a study of acute UC mice, oral sodium butyrate was able to reduce inflammation and mucosal damage [53]. In another study, SCFA administration in mice decreased the incidence and size of tumors in colitis-associated colorectal cancer induced by azoxymethane (AOM) and DSS [54].

Bile acids are another class of important metabolites in FMT. Approximately 95% of bile acids are reabsorbed through enterohepatic circulation. The deconjugated effect of gut microbes on bile acids prevents some of them from being reabsorbed by the apical sodium-dependent bile acid transporter (ASBT) in the ileum [55]. Unabsorbed primary bile acids (cholic acid, CA; chenodeoxycholic acid, CDCA) enter the colon and undergo 7-dehydroxylation metabolism to transform into secondary bile acids (deoxycholic acid, DCA; lithocholic acid, LCA; ursodeoxycholic acid, UDCA) [55]. In an in vitro experiment, primary bile acids were found to promote the germination of *Clostridioides difficile*, while secondary bile acids restricted its germination, growth, and toxin activity [56]. In a mouse experiment, oral UDCA downregulated the severity of DSS-induced colitis [57].

### 2.3. Immunoglobulin A

Immunoglobulin A (IgA), the first line of defense for intestinal epithelium, protects host cells against pathogens and toxins in the gut. In Grellet et al.’s study [58], puppies with intestinal pathogen shedding were found to have lower fecal IgA concentrations than those without shedding. Therefore, the timely replenishment of IgA is very important for improving disease treatment. For example, oral W27 IgA has been found to have a good effect on lymphoproliferative disease and colitis in mice models [59]. Vancomycin-mixed IgA treatment has been found to improve CDI hamster survival compared to treatment with vancomycin alone [60]. FMT might introduce IgA from donor feces into the intestines of diseased dogs to aid in the intestines’ function.

## 3. The Mechanism of Important Components Acting in FMT

### 3.1. Affecting the Microbiota

Canine gastrointestinal diseases may cause microbiotic dysbiosis. Therefore, gut microbiota restoration, especially bacterial recovery, is the most critical step in disease treatment. FMT is a fast and effective method for the alteration of the bacteria in the recipient’s intestines. In CDI, antibiotic-induced bacterial dysbiosis is regarded as the trigger of toxigenic *Clostridioides difficile* colonization, which makes antibiotic therapy inapplicable [61]. Under this situation, FMT can increase intestinal microbial diversity and richness and eliminate toxigenic *Clostridioides difficile*, allowing for recovery from canine CDI [4]. In dogs with CPV enteritis, FMT has great potential to aid in the recovery of gut microbiota, producing more abundant *Proteobacteria* and less *Bacteroidetes* [62]. Further, for FMT in dogs with IBD, *Fusobacteria* increased from 0 to 35%, while *Proteobacteria* decreased from 52.2% to 1.5%, a level that is similar to that of healthy dogs [5]. Significant microbiota changes were also seen in dogs with AHDS, with increases in the Shannon diversity index, *Clostridium hiranonis,* and SCFA-producing bacteria (*Eubacterium biforme, Faecalibacterium prausnitzii, and Prevotella copri*) observed [10].

Similarly, viruses and fungi also return to normal levels due to FMT. Research has found that transplanting a sterile fecal filtrate results in substantial changes in phages, i.e., to a state similar to that of the donor after 6 weeks [12]. Patients with CDI have been found to have a higher abundance and a lower diversity, richness, and evenness of *Caudovirales* viruses in their intestines; FMT may be able to alter this [44,63]. In fungi, *Candida* and *Saccharomyces* are abundant commensal fungal classes in humans, mice, and dogs [20,64]. In a study, the ratio of *Basidiomycota*/*Ascomycota* and *Candida albicans* increased while the component of *Saccharomyces cerevisiae* decreased in mice with IBD compared to healthy mice [65]. FMT may improve this condition, as a study showed that *Candida* levels were high before FMT but dropped after FMT in human UC [66]. Zuo et al. [67] also found that *Saccharomyces* and *Aspergillus* increased in mice with CDI after FMT treatment.

Along with the alterations of microbiota after FMT, bacterial metabolites (SCFAs, bile acids, and other beneficial components) change correspondingly. For example, after FMT treatment in dogs with AHDS, the abundance of the SCFA-producing bacteria increased significantly, which means that SCFAs increased accordingly [10]. Similar changes in SCFAs have been reported in the intestines of human patients with recurrent CDI after FMT treatment [68,69]. Further, the transformation of primary bile acids to secondary bile acids is affected by bacteria with bile acid-inducible enzymes [70]. In a study, the feces of dogs with AD were found to have high primary bile acid levels and low secondary bile acid levels. After FMT therapy, the primary bile acid levels in the feces significantly decreased, while the secondary bile acid levels showed an upward trend [9].

#### 3.1.1. Commensal Bacteria, Viruses, and Fungi

Commensal bacteria from donor feces compete with pathogens for living space, nutrients, and other resources to inhibit the growth of pathogens in the recipient’s intestinal tract. For example, NTCD strains more easily adhere to intestinal epithelial cells (IECs) than toxigenic *Clostridioides difficile* strains, thus restricting pathogen colonization [28]. In addition, commensal *Enterobacteriaceae* in the intestine have been found to generate colonization resistance to *Salmonella* through oxygen competition [71]. *Lactobacillus reuteri* may secrete a mucus-binding protein, which reduces the adhesion of *Clostridioides difficile* [72,73]. Commensal *Escherichia coli* (*E. coli)* has been found to consume organic acids, amino acids, and other nutrients that are also needed by enterohaemorrhagic *E. coli* (EHEC) [74]. Additionally, commensal bacteria secrete molecules to directly impact pathogenic bacteria. For instance, commensal *E. coli* has been found to secrete bacteriocins that suppress the growth of EHEC [74]. By generating the hypoxia-inducible factor (HIF)-1α and the antimicrobial peptide (AMP) LL-37, commensal bacteria are capable of restricting *Candida albicans* colonization, which is thought to be related to CDI [75]. SCFAs, the metabolites of commensal bacteria, may downregulate the virulence genes of *Salmonella enterica* serovars *Enteritidis* and *Typhimurium* [74]. Secondary bile salts synthesized by a few commensal bacteria have been found to have an inhibitory effect on *Clostridioides difficile* [76,77]. In summary, through competing with pathogenic bacteria for living resources and through secreting inhibitory molecules, commensal bacteria from donor feces may curb the growth of pathogenic bacteria directly.

Phages and fungi from donor feces are also considered to be important in FMT. Beneficial phages can optimize the structure and composition of their host microbiota through changing the virulence of bacteria. Almost every structure on the surface of bacteria (including lipopolysaccharides, outer membrane proteins, peptidoglycan, etc.) can be used as a phage receptor [78]. Phages can also modify virulence structures to reduce the virulence of bacteria [78]. Further, phages can alter bacterial antigenicity by producing enzymes [39]. With regard to fungi, the interaction of fungal microflora and bacterial microflora in the intestine may affect the occurrence and development of diseases [65]. For example, *Saccharomyces boulardii* produces the 54 kDa serine protease hydrolysising toxin A of *Clostridioides difficile* and its receptor [79].

#### 3.1.2. Bile Acids and IgA

Bile acids are important nutrients for many beneficial bacteria that can promote the growth of bile acid metabolizing bacteria and inhibit the growth of other bile-sensitive bacteria. For example, studies have illustrated that secondary bile acids increase *Akkermansia muciniphila* abundance and limit *Clostridium* cluster XIVa loss [57,80]. Bile acids possess a bactericidal action, as they have a deterrent effect on bacterial cell membranes [81]. Further, they possess the ability to cause DNA damage and oxidative damage [70,81]. Okai et al. [59] found that high-affinity polyreactive W27 IgA regulated the microbiota in mice intestines by binding to colitogenic bacteria instead of beneficial bacteria. Furthermore, IgA has the capacity to promote commensal bacteria to adhere to epithelial cells, enhancing the colonization of commensal bacteria [82,83,84,85].

### 3.2. Maintaining the Intestinal Barrier

Canine intestinal diseases are usually accompanied by damage to the intestinal barrier. The intestinal barrier includes bacteria, mucus, IECs, gut-associated lymphoid tissue, and immune cells [86]. The mucus layer is divided into the outer mucus layer and the inner mucus layer. The outer mucus layer is the main area where microbial colonization occurs [76,86]. The inner mucus layer is a dense network layer formed mainly by mucin 2 (MUC2) polymers, which are produced by goblet cells [76,87]. In a study, the expression levels of MUC2 in the intestines of humans with CDI were found to be lower than those in healthy people. Further, the number of goblet cells in people with UC’s ilea were found to be reduced significantly [76]. The inner mucus layer, which contains AMPs and IgA, separates microorganisms from the intestinal epithelia [76,88]. The connection among IECs is regulated by apical junctional complexes (AJC). Tight junctions (TJs) play a key role in the epithelial barrier and mucosal permeability [89,90]. Important proteins contributing to TJs belong to the claudin family, including occludins, tricellulin, zonula occludens, (ZO)-1, ZO-2, and ZO-3, cingulin, etc [91]. When intestinal diseases occur, these barrier structures are impaired. In dogs with IBD, goblet cells were found to be lost, and TJ strands were clearly reduced [92]. The *Clostridioides difficile* toxin is able to destroy colonic epithelial cells, exposing the host to gut microbes [93]. The necrosis of intestinal epithelia, which is caused by the *Clostridium perfringens* toxin, is an important histopathologic lesion in dogs with AHDS [94]. Similarly, in CPV-infected dogs, the replication of the virus in IECs can damage the intestinal barrier [95]. Hence, the restoration of the intestinal barrier after FMT is particularly important. A mouse study indicated that FMT may enhance intestinal barrier restoration through decreasing epithelial cell apoptosis, adjusting the mucus layer, and upregulating TJ proteins [96,97,98].

#### 3.2.1. Bacteria

The interaction between bacteria and the intestinal barrier mainly depends on the pattern recognition receptors’ (PRRs) recognition of microbe-associated molecular patterns (MAMPs), such as lipopolysaccharides, flagellin, bacterial DNA, and RNA. MAMPs’ binding to PRRs, such as toll-like receptors (TLRs) and NOD-like receptors (NLR), activates downstream adaptor molecules, stimulating a series of immune responses [99].

The benefits of commensal bacteria from the donor’s feces in maintaining the recipient’s intestinal barrier function are characterized by two main aspects. On the one hand, these bacteria are beneficial to the restoration of the intestinal barrier structure. Transplanted probiotics have been found to be able to improve intestinal barrier function by regulating the expression of junction complexes [76]. For example, *Bifidobacterium bifidum* and *Faecalibacterium prausnitzii* have been found to strengthen apical junctions, reducing intestinal epithelial permeability [100]. *Akkermansia muciniphila* has been found to increase TJ (occludins, claudins, and ZO-1/2/3) expressions [101]. Additionally, transplanted commensal microorganisms may participate in the intestinal lymphoid structure’s development, promote the maturation of IECs, and accelerate the angiogenesis of intestinal mucosa [102]. For example, *Akkermansia muciniphila* and *Faecalibacterium prausnitzii* have been found to impact the number and development of goblet cells [100]. By promoting goblet cells’ secretion of mucins, transplanted commensal bacteria are beneficial for the reconstruction of the mucus layer in the intestine [76]. Bacteria can produce butyric acids to promote the release of mucin [88].

On the other hand, transplanted commensal bacteria also stimulate the host to secrete IgA antibodies and AMPs (such as Reg-IIIγ, α-defensins, and β-defensins), adjusting the composition of the mucus layer in order to resist pathogens. Commensal bacteria can stimulate the host to release IgA through the TLR pathway, which forms the first line of defense [103]. In addition to IgA, Brandl et al. [104] found that the synthesis and secretion of Reg-IIIγ are induced by commensal bacteria through TLR receptors and likely not by pathogenic bacteria. *Akkermansia muciniphila* is a commensal bacterium that produces Reg-IIIγ [101]. Commensal bacteria have been found to regulate the DefA gene of Paneth cells through TLR-MyD88 signal transduction in order to affect the secretion of α-defensin [105], which has been proven to only eliminate pathogenic bacteria, not commensal bacteria [106]. Intestinal probiotics have been found to induce Caco-2 epithelial cells to produce β-defensins through TLRs [107]. In addition to IgA and AMPs, angiogenin 4 (Ang4) also has the ability to prevent pathogenic microorganisms from entering into the intestinal epithelium and hindering the inflammatory response [108]. Ang4 is generated when commensal bacteria come into contact with intestinal mucosa. *Bacteroides thetaiotaomicron* has been proven to raise the expression of Ang4 [108].

#### 3.2.2. SCFAs and Bile Acids

SCFAs play irreplaceable roles in repairing the intestinal barrier. To influence the host, SCFAs inhibit histone deacetylase (HDAC) and bind to the corresponding receptors, such as the SCFA-sensing G protein-coupled receptor (GPCR), also known as the free fatty acid receptor (FFAR), which includes GPR41 (FFAR3), GPR43 (FFAR2), and GPR109 (hydroxy-carboxylic acid receptor 2 or HCA2) [109,110]. Since GPCRs are expressed in almost all immune cells (such as IECs, neutrophils, and macrophages), SCFAs can induce immune responses by activating GPCRs. SCFAs regulate IECs mainly through GPR43 and GPR109a and act as regulators of mucin, AMPs, IgA, chemokines, and cytokines [109].

SCFAs protect the intestinal barrier through multiple methods. (1) SCFAs (mainly acetates) are components of intestinal mucosal nutrients, which provide 60%-70% of the energy used in the metabolism of intestinal mucosal cells [50,111,112]. (2) By regulating transcription HIF, SCFAs maintain the anaerobic environment in the colon, which is beneficial to commensal anaerobe bacteria [50,111,113]. (3) SCFAs promote intestinal mucosal hyperplasia and intestinal mucus secretion to maintain the height, width, recess depth, and mucosal thickness of intestinal villi, increase the proliferation capacity of intestinal mucosal cells, and reduce intestinal mucosal atrophy caused by inflammation [114,115]. (4) SCFAs stimulate the synthesis of the intestinal mucosal TJ protein ZO-1 and occludin-5, strengthen intestinal mucosal barrier function, and reduce the entry of harmful substances (such as lipopolysaccharides) into the blood [115,116,117]. (5) SCFAs enhance intestinal barrier function also by increasing AMP and IgA secretion [116,118]. (6) Furthermore, SCFAs regulate NLRP3 through GPR43 and GPR109a on IECs to promote the immune response and mucosal protection [119]. SCFAs’ binding to receptors causes K^+^ efflux, Ca2^+^ flux, and hyperpolarization, leading to NLRP3 inflammasome activation. The activated NLRP3 inflammasome stimulates caspase-1 into its active form, then converts pro-IL-18 into IL-18. IL-18 is helpful for limiting mucosal damage and preventing the activation of immune cells in the mucosal lamina propria, which is essential for maintaining epithelial integrity and intestinal homeostasis [119].

In addition to SCFAs, the physiological levels of bile acids can also aid the intestinal barrier through inducing goblet cells’ secretion of mucins, stimulating cell migrations, and mediating cytokine secretions in innate immunity, as well as advancing the expression of AMPs [120].

### 3.3. Anti-Inflammation

Bacterial components and inflammatory mediators mediate cytokine production mainly through the nuclear factor-κB (NF-κB) and the mitogen-activated protein kinase (MAPK) signaling pathway [121]. Bacterial components trigger NF-κB and MAPK through the PRRs of innate immune cells. Intestinal innate immune cells include epithelial cells, macrophages, dendritic cells (DC), mast cells, eosinophils, natural killer (NK) cells, mesenchymal cells, endothelial cells, etc. The intestinal mucosa also contains T helper 17 (Th17) cells and Foxp3+ regulatory T cells (Tregs), both of which derive from CD4+ T cells. Th17 cells produce IL-17, IL-22, and IL-23, which cause inflammation, while Treg cells produce IL-10 and TGF-β, which suppress inflammation [122]. IL-10 downregulates the transmission of proinflammatory signaling, slows down the response of Th17, and avoids excessive immune damage during inflammation [102].

Inflammation is usually found in canine intestinal diseases, such as CDI, IBD, CPV enteritis, AD, and AHDS. In CDI, the levels of proinflammatory mediators, such as CXCL5, IL-8, IL-23, and IFN-γ, have been found to be higher [123]. Similarly, in IBD, which is characterized by the inflammatory infiltrate, the NF-κB signaling pathway has been found to be upregulated and the production of IL-23 increased. IL-23 has been shown to stimulate the differentiation of CD4+ T cells that produce IL-17 [92]. In CPV infection, enteritis, myocarditis, and systemic inflammatory response syndrome (SIRS) may take place in dogs. Consequently, controlling the occurrence of inflammation and secondary infection is of great importance in CPV treatment. Similarly, enteritis as well as SIRS may appear in AD and AHDS [124,125]. Thus, anti-inflammation is an important focus in the treatment of intestinal diseases. Research on humans has shown that FMT may inhibit intestinal inflammation [126]. After FMT was applied to mice with intestinal inflammation, the levels of the proinflammatory cytokines IL-1β, IFN-γ, and TNF were found to have decreased, while the level of the anti-inflammatory cytokine IL-10 was found to have increased [127,128,129].

#### 3.3.1. Commensal Bacterial, Viruses and Fungi

Commensal bacteria in FMT inhibit inflammation in a variety of ways. *Lactobacillus acidophilus* in dogs attenuates LPS-induced IL-8 secretion in vitro [26]. *Akkermansia muciniphila* mediates Treg proliferation [101]. *Faecalibacterium prausnitzii* transcytosis enables it to interact with TLRs, NLRs, and C-type lectin receptors (CLRs) on DCs and then induces Tregs [130]. Further, commensal microorganisms from donor feces exert an anti-inflammatory function through generating immunomodulatory molecules. By producing SCFAs, commensal *Clostridium* induces Treg production in the gut [122]. The polysaccharide A (PSA) from *Bacteroides fragilis* has been found to bind to the TLR2 of T-cells, inducing the production and function of IL-10 and Tregs, while limiting the response of Th17 [99,122]. *Faecalibacterium prausnitzii* secrets microbial anti-inflammatory molecules (MAMs), inhibiting the activation of NF-κB and the secretion of IL-8; promoting CD103+’s migration to the mesenteric lymph nodes, thus inducing the production of Tregs; and stimulating antigen-presenting cells to produce IL-10, which increases Foxp3+ Treg activity and blocks the function of Th17 cells [130]. *Faecalibacterium prausnitzii* also secrets SCFAs to suppress NF-kB activation [130]. *Bacteroides thetaiotaomicron*, the most abundant bacteria in the intestinal tract, facilitates the nuclear export of the RelA subunit in NF-κB, thereby antagonizing the NF-κB transcription factor for anti-inflammatory purposes [131].

The viruses from donor feces, including phages, maintain immunological homeostasis as well [39]. Phages can enter into the organization and the circulatory system, then come into contact with immune cells to trigger immune-related responses [39]. Studies have shown that phages reduce TNF-α, IL-1β, and IL-6 in the blood of mice with *Pseudomonas aeruginosa*-induced sepsis [42]. In a study, the administration of the phage tail adhesion protein (gp12) eliminated almost all of the IL-1α and reduced half of the IL-6 in LPS-injected mouse serum. This protein also decreased leukocytic infiltration in the lungs, spleen, and liver [132]. The T4 phage controls the production of reactive oxygen species (ROS), thus suppressing the immune response to inflammation [133].

In addition to commensal bacteria and viruses, fungi also can curb inflammation. *Saccharomyces cerevisiae* has been found to have an anti-inflammatory effect through mediating the expression of IL-10, which can discourage colitis caused by adherent-invasive *E. coli* [65]. *Saccharomyces cerevisiae* has been found to cut down proinflammatory cytokine levels and to impact the activation of NF-κB, mitogen-activated protein kinases p38 and JNK, and AP-1 [45].

#### 3.3.2. SCFAs and Bile Acids

It is well known that SCFAs themselves can regulate host immunity and support immune homeostasis. When intervening in UC, SCFAs decrease proinflammatory cytokine secretions and increase anti-inflammatory cytokine secretions, mediating anti-inflammation [50]. SCFAs can suppress NF-kB activity through HDAC inhibition [115]. SCFAs have been proven to suppress LPS-induced proinflammatory cytokines (such as IL-6 and IL-12p40) and promote the production of the anti-inflammatory cytokine IL-10 [134]. SCFAs also induce Treg development controlled by the forkhead box P3 (Foxp3) promoter to exert anti-inflammatory effects. Butyrate and propionate directly interact with naive T cells, raising the acetylation of the transcription factor Foxp3 promoter through inhibiting HDAC and promoting the differentiation of Tregs [111,115]. GPR43 on colonic T cells also increases the expression of Foxp3 [109]. Further, the combination of SCFAs and GPR109a stimulates DCs and macrophages to release IL-10 and Aldh1a, inducing Treg generation [109].

Secondary bile acids limit Caco-2 cells’ secretion of IL-8 after IL-1β stimulation, while primary bile acids do not have this function [70]. Moreover, secondary bile acids can accelerate Treg differentiation and restrain Th17 differentiation [70]. Important bile acid activated receptors are the farnesoid X receptor (FXR), the G protein-coupled bile acid receptor 1 (GPBAR1), the pregnane X receptor (PXR), and the vitamin D receptor (VDR) [55,70,135]. There is evidence that bile acids can induce the transcription of host antibacterial agents to carry out indirect bactericidal actions through FXR [136]. Furthermore, the activation of FXR inhibits epithelial permeability, reduces the loss of goblet cells, and suppresses the production of proinflammatory cytokines (such as the NF-κB-dependent cytokines IL-6, IL-1β, and TNF-α) in different immune cell populations [137]. GPBAR1 is highly expressed in monocytes/macrophages, and the main ligands of GPBAR1 are secondary bile acids [55,138]. GPBAR1 promotes the transformation of classically activated (M1) macrophages (high IL-12 and low IL-10-producing) into alternatively activated (M2) macrophages (low IL-12 and high IL-10-producing) in the intestine [138,139,140]. M2 macrophages have an anti-inflammatory function and may be important in the field of IBD treatment [138]. The activation of GPBAR1, PXR, and VDR is likely to curb NF-κB, further limiting inflammation or IEC impairment [70]. The conversion of primary bile acids into secondary bile acids occurs in the colon. To activate the receptors in the small intestine, secondary bile acids must be reabsorbed in the colon and secreted again into the small intestine [55,141]. FMT through upper gastrointestinal administration seems to be able to achieve this effect.

## 4. Discussions

Different components play different roles in the intestine. Evidence has shown that the efficacy of probiotics is strain-specific and disease-specific [142]. In other words, the mechanisms of commensal bacteria active in the intestine vary for different strains as well as for different diseases. Similarly, different strains of viruses and fungi in the intestine also perform differently. From the above review, the main ways these components exert functions in FMT may be summarized as follows: affecting the microbiota (Figure 1), maintaining the intestinal barrier (Figure 2), and inducing anti-inflammation (Figure 3) in the recipient’s intestines.

### 4.1. Other Components in Feces May Be Functional in FMT

Generally, only the effective components in FMT are determined, and damage to these components should be avoided as much as possible during the preparation and preservation of fecal material; further, the correct perfusion method should also be chosen. Beyond the components discussed above, mucin proteins and IECs as well as other fecal constituents are likely to be functional ingredients in FMT. Highly glycosylated mucin proteins are carbon and energy sources for intestinal microbiota [143]. Various gut anaerobic bacteria possess enzymes that degrade mucin oligosaccharides into monosaccharides, (such as N-acetyl-D-glucosamine, N-acetylgalactosamine, galactose, fucose, and sialic acid), which are nutrition sources for microbes [87,143]. The co-transplantation of mucins and bacteria might provide a better environment for bacteria in the process of FMT and colonization. Moreover, introducing mucins as a component of the intestinal barrier may protect the recipient’s intestinal epithelium from pathogens. Transplanting colonic stem cells into colon-damaged mice has been found to allow the areas lacking colon cells to be covered by epithelium. Bojanova et al. [11] suggested that colonocytes may serve as effective components in FMT if colonic stem cells can be isolated from feces. In addition, bacterial fragments, such as bacterial cell wall components and DNA fragments, stimulate the host to respond to PRRs, thereby regulating the ecological niche of commensal bacteria [12].

### 4.2. FMT May Be Effective in Many Other Diseases of Dogs

Due to the multiple functions of FMT components, these components not only have beneficial effects on gastrointestinal disorders (such as CDI, IBD, irritable bowel syndrome, idiopathic constipation, limiting antibiotic-resistant bacterial infections, and pouchitis), but also improve metabolic disorders (such as metabolic syndrome, obesity, nonalcoholic fatty liver disease, diabetes, myocarditis, and vascular inflammation), neuropsychiatric disorders (such as hepatic encephalopathy, Parkinson’s disease, autism spectrum disorder, and multiple sclerosis), and immunologic disorders (such as rheumatoid arthritis, graft-versus-host disease, and colorectal cancer) in humans [1,2,61,144,145]. Intestinal barrier and microbiota restorations play important roles in FMT when treating extraintestinal diseases since they are associated with intestinal microbial dysbiosis. The normalization of the microbiota and metabolites in the intestine further impacts host immunity and metabolism. For instance, SCFAs have been found to promote the host to secrete glucagon-like peptide-1, which can lower the serum levels of glucose, increase insulin secretion and resistance, and protect pancreatic β-cell function in the treatment of diabetes [146]. Bile acids can regulate glucose homeostasis, insulin sensitivity, and energy metabolism [147]. The anti-inflammation effects of FMT have been observed in many diseases, such as diabetes, nonalcoholic fatty liver disease, myocarditis, vascular inflammation, colorectal cancer, rheumatoid arthritis, and graft-versus-host disease, which are all characterized by inflammation.

For dogs, the positive effects of FMT on CDI, IBD, CPV enteritis, AD, and AHDS have been found, but more studies are still needed to support the current findings. Drawing on indications of FMT in humans, it is hypothesized that FMT may also have positive impacts on some similar diseases, such as limiting antibiotic-resistant bacterial infections, obesity, diabetes, myocarditis, vascular inflammation, hepatic encephalopathy, rheumatoid arthritis, and colorectal cancer, since they also occur in dogs [148,149,150,151,152,153,154]. Similarly, these hypotheses still need to be proven experimentally.

### 4.3. FMT in Veterinary Clinics

In veterinary clinics, the use of FMT is thought to be a positive prospect for some diseases but also to have certain challenges. The growth of antibiotic resistant bacteria has always been a concern in medicine. By utilizing microbes and other components in feces to influence the gut and the body’s immune system, FMT has become an alternative to antibiotic therapy to some extent. It has been proven to improve the treatment of certain diseases (such as IBD), which traditional drug therapies have failed in [6,7]. However, great issues regarding its clinical application need to be solved. To avoid the co-transfer of dangerous factors (such as infectious diseases), rigorous screening should be carried out on donor feces. Unlike in human FMT, “centralized stool banks” are not set up for dogs [61], which makes canine FMT complicated and expensive since every canine recipient needs to find a donor, and the donor needs to go through a thorough screening process. Building up “stool banks” seems to be necessary, but has some challenges. First, FMT is not widely used in veterinary clinics. Second, long-time conservation will gradually deactivate FMT’s functional ingredients. Due to the low-frequency use of FMT in dogs, it is possible that fecal materials will be stored until they are inactive and will still not be used. In addition, research on canine or pet FMT is limited, and details about its clinical application are ambiguous. For example, since the body types and breeds of dogs are diverse, whether one donor’s fecal material can be used in breeds different from it is still unknown. Further, how to calculate the dosages of fecal material for canine recipients with different body types is still to be determined. The frequency and duration of fecal material administration for different indications are also unclear. More research on canine FMT is therefore needed.

## 5. Conclusions

In conclusion, it is generally believed that the main components that make FMT effective are commensal bacteria in feces. Nonetheless, other components in feces, such as viruses, fungi, immunoglobulin, and bacterial metabolites, also play important roles in canine FMT. It is important to preserve these components as much as possible in the preparation of fecal material. Moreover, donors can be given a high-fiber diet for a period of time before collecting their feces since a high-fiber diet in dogs favors bacterial biodiversity and is the core source of SCFAs [155]. It is worth noting that in FMT, oral SCFAs will be absorbed and oxidized rapidly, and enemas or colonoscopies can solve this problem. When we previously discussed the indications of canine FMT, enemas were reported as the most commonly used fecal transplant method in dogs, which may be partly due to the protection of SCFAs [3]. On the contrary, oral delivery of FMT can provide an opportunity for bacteria to colonize in the small intestine and ileum. Meanwhile, oral administration also allows metabolites, which are produced in the colon (such as secondary bile acids), to enter the small intestine and ileum where they can function. In consequence, oral combined enema/colonoscopy administration is thought to be a better method for canine FMT, which is supported by Bottero et al. [6]. Additionally, as viruses cannot survive without cells, the preservation of cellular components (such as bacteria, archaea, fungi, and IECs) in fecal material is important.

## Figures and Tables

**Figure 1 vetsci-09-00695-f001:**
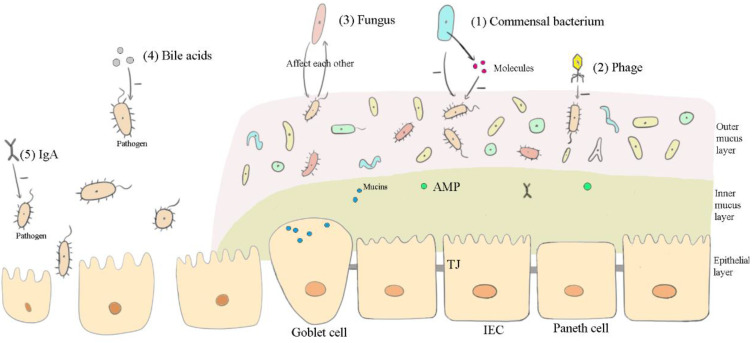
How the important components in FMT affect the microbiota of the recipient’s intestines. “+” represents promotional effects, while “-” represents suppressive effects. (1) Commensal bacteria from donor feces compete with pathogenetic bacteria for resources to inhibit their growth. Commensal bacteria also secrete molecules (such as bacteriocins) to directly impact pathogenic bacteria. (2) Phages, through lysing bacteria or modifying bacterial structures, kill bacteria or alter their virulence. (3) Fungal microflora and bacterial microflora in the intestine impact one another. (4) Bile acids promote the growth of bile acid-metabolizing bacteria and inhibit the growth of other bile-sensitive bacteria. (5) Immunoglobulin A (IgA), as the first line of defense, suppresses pathogens and their toxins in the gut.

**Figure 2 vetsci-09-00695-f002:**
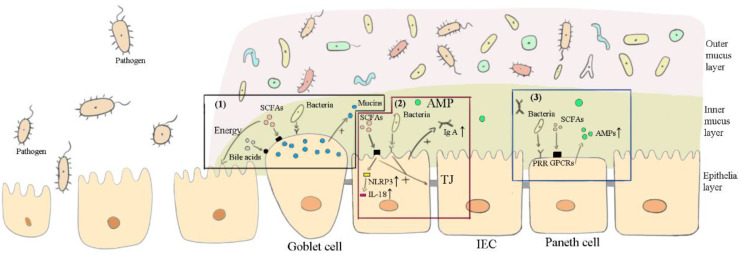
How the important components in FMT maintain the intestinal barrier. “+” represents promotional effects, while “-” represents suppressive effects. (1) Commensal bacteria, short-chain fatty acids (SFAs), and bile acids bind to their corresponding receptors in goblet cells, mediating mucin secretion. SCFAs provide the main energy for epithelial cells (IECs). (2) Commensal bacteria and SFAs upregulate tight junction (TJ) expressions in intestinal IECs and promote the secretion of IgA. SCFAs increase NLRP3 inflammasomes through G protein-coupled receptors (GPCRs) and further convert pro-interleukin-18 (pro-IL-18) into IL-18, which is beneficial for the intestinal barrier. (3) Commensal bacteria and SFAs promote Paneth cells to secrete antimicrobial peptides (AMPs).

**Figure 3 vetsci-09-00695-f003:**
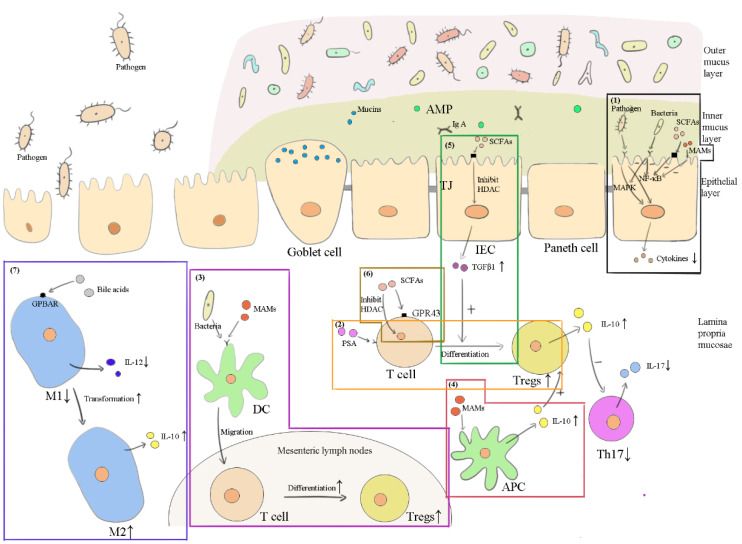
How the important components in FMT exert anti-inflammatory actions. “+” represents promotional effects, while “-” represents suppressive effects. (1) Commensal bacteria inhibit the nuclear factor-κB (NF-κB) and the mitogen-activated protein kinase (MAPK) signaling pathway to decrease inflammatory cytokines. SCFAs (from donor feces or bacteria) and microbial anti-inflammatory molecules (MAMs from bacteria) inhibit NF-κB. (2) Polysaccharide A (PSA) from bacteria binds to the toll-like receptor (TLR) 2 of T cells, inducing regulatory T cell (Treg) production. (3) MAMs and bacteria stimulate dendritic cells (DCs) to migrate to the mesenteric lymph nodes, thus inducing Treg production. (4) MAMs also stimulate antigen-presenting cells (APCs) to produce IL-10. (5) SCFAs promote IECs to secrete the transforming growth factor (TGF) β1, which boosts the differentiation of T cells into Tregs. (6) SCFAs can directly induce Treg production through GPR43 on T cells or by inhibiting histone deacetylase (HDAC) in T cells. (7) Bile acids promote classically activated (M1) macrophages’ transformation into alternatively activated (M2) macrophages to limit inflammation.

## Data Availability

Not applicable.

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
