# Peer review of "The Mechanism of Important Components in Canine Fecal Microbiota Transplantation"

_vetsci, 2022, doi:10.3390/vetsci9120695_

Round 1

Reviewer 1 Report

Dear authors,

Congratulations on your work which is an excellent contribution to the understanding of the role of fecal transplants in veterinary gastroenterology. Then I leave my contribution to make the work easier to read.

Abstract

intestinal epithelial cells (IECs)

Introduction

 As you write later in the text: In CPV infection, enteritis, myocarditis, and systemic inflammatory response syndrome (SIRS) may take place in dogs. Then please change "Canine parvovirus (CPV) diarrhea" by canine parvovirus (CPV) enteritis....

2.2. Metabolites

The metabolites thought to be most potent in FMT are short chain fatty acids (SCFA) or postbiotics

3.3. Anti-inflammation

3.3.1. Commensal bacterial, viruses and fungi

Commensal bacteria in FMT inhibit inflammation in a variety of ways.

Bacteroides thetaiotaomicron instead "Thetaiotamicron, the abundant bacteria in intestinal tract..."

4.2. FMT may be effective in many other diseases of dogs

This point concerns the dog but all references pertain to human medicine. However, the distinction between human and dog pathology is never really clear to the reader. Please clearly distinguish the two situations.

ex: multiple sclerosis, rheumatoid arthritis, and colorectal cancer in dogs ????

Is "antibiotic-resistant bacteria" a gastrointestinal disorder?

Figure 1 - "(5) Immunoglobulin A (IgA) as the first line of defense suppresses pathogens and their toxins in the gut." IgA is not represented in this figure.

Reviewer 2 Report

The Mechanism of Important Components in Canine FMT

FMT is a reemerging therapeutic method extensive research is carried out. It’s use in humans as well as animals can help to treat several diseases for which we do not have an effective therapy. There are several diseases for which we need to try FMT in dogs. This review addresses that. The figures are excellent. I have some comments below, please kindly address them.

 Clostridium difficile = Clostridioides difficile

2.1. Gut microorganisms mainly contain archaea, fungi, bacteria, virus, and eukaryotes. Fungi are also eukaryotes; are you referring here to protozoa?

3.1. “For example, it has been regarding as the core risk factor for CDI that antibiotic-induced bacterial dysbiosis, influencing the colonization resistance to toxigenic Clostridium difficile.” A convoluted statement, please rephrase it for more clarity.

3.1. “What's more, after FMT was used in IBD dogs, Fusobacteria increased from 0 to 35% while Proteobacteria de-creased from 52.2% to 1.5%, towards a level that was similar to healthy dogs.[3].”

3.1.2 “Furthermore, IgA has the capacity to promote commensal bacteria to adhere to epithelial cells enhancing the colonization of commensal bacteria[61].” Please cite more publications to strengthen the claim.

23.3.1. Commensal bacterial, viruses and fungi”

4. “For different components, their roles in intestines are different.” What do “their” refer to?

I would suggest to briefly address the following points

List of diseases we may potential use FMT for in dogs or pet in general.

What are the challenges of using FMT in dogs? Practical, ethical,..issues, AMR transfer and others

Prospective of FMT, what would need to be addressed to use FMT widely in vet clinics/hospitals?

I would suggest two or three research articles to support your claim; nearly in all places, you cited only one article.

Reviewer 3 Report

The most commonly used fecal transplant methods in the dog could be added

Round 2

Reviewer 2 Report

Thank you for addressing my comments and answering my questions. I would suggest to thoroughly review the manuscript for language.

Author Response

Thanks for the comment of improving English language. We have reviewed the manuscript again for language.